# Tubulin tyrosination/detyrosination regulate the affinity and sorting of intraflagellar transport trains on axonemal microtubule doublets

Aditya Chhatre [1,2,3], Ludek Stepanek [2,4], Adrian Pascal Nievergelt [2], Gonzalo Alvarez Viar[5], Stefan Diez [1,2,3] ✉ & Gaia Pigino [1,2,5] ✉

Cilia assembly and function rely on the bidirectional transport of components between the cell body and ciliary tip via Intraflagellar Transport (IFT) trains. Anterograde and retrograde IFT trains travel along the B- and A-tubules of microtubule doublets, respectively, ensuring smooth traffic flow. However, the mechanism underlying this segregation remains unclear. Here, we test whether tubulin detyrosination (enriched on B-tubules) and tyrosination (enriched on A-tubules) have a role in IFT logistics. We report that knockout of tubulin detyrosinase VashL in *Chlamydomonas reinhardtii* causes frequent IFT train stoppages and impaired ciliary growth. By reconstituting IFT train motility on de-membranated axonemes and synthetic microtubules, we show that anterograde and retrograde trains preferentially associate with detyrosinated and tyrosinated microtubules, respectively. We propose that tubulin tyrosination/detyrosination is crucial for spatial segregation and collision-free IFT train motion, highlighting the significance of the tubulin code in ciliary transport.

Cilia (or eukaryotic flagella) are conserved microtubule-based organelles of eukaryotic cells that play fundamental roles in fluid motility, cell swimming, and signalling. For their assembly and maintenance, they require specialized molecular bi-directional transport machines, the intraflagellar transport (IFT) trains[1]. IFT trains consist of large polymeric assemblies of two large protein complexes known as IFT-A (6 proteins) and IFT-B (16 proteins)[2]. They are powered by microtubule-based molecular motors. Specifically, kinesin-2 and dynein-1b (dynein-2 in humans), respectively, drive the anterograde (base-to-tip) and retrograde (tip-to-base) IFT trains. Thereby, the motion of each IFT train is restricted to a very narrow space just underneath the ciliary membrane along the external face of the nine microtubule doublets that compose the ciliary axonemal structure[1]. Despite the restricted and crowded environment, collisions between oppositely directed IFT trains and consequent traffic jams are not observed in motile cilia. This is because opposite polarity trains sort onto distinct tubules of the microtubule doublets[3], effectively clearing the way for oncoming trains. Retrograde trains move on the A-tubule, while anterograde trains on the B-tubule of the doublet[3]. However, the mechanism responsible for the sorting of the IFT trains onto the different tubules remains elusive.

Though being composed of different numbers of protofilaments, A-tubules and B-tubules have the same tubulin-lattice structure[4]. They are made of the same tubulin isoforms but have been shown to be

[1]Cluster of Excellence Physics of Life, TUD Dresden University of Technology, 01062 Dresden, Germany. [2]Max Planck Institute of Molecular Cell Biology and Genetics, 01307 Dresden, Germany. [3]B CUBE - Center for Molecular Bioengineering, TUD Dresden University of Technology, 01307 Dresden, Germany. [4]Institute of Molecular Genetics, Czech Academy of Sciences, 14220 Prague, Czech Republic. [5]Human Technopole, 20017 Milan, Italy. ✉e-mail: stefan.diez@tu-dresden.de; gaia.pigino@fht.org

differentially enriched in certain tubulin post-translational modifications[5]. Thus, it has been speculated that tubulin post-translational modifications might encode different biochemical cues on the surface of the two tubules that contribute to the spatial segregation of anterograde and retrograde IFT trains. Tubulin polyglutamylation was shown to be enriched in B-tubules[5,6] and we have recently shown that this post-translational modification specifically marks only a single tubulin protofilament (B9)[7]. Polyglutamylation contributes to the regulation of the Nexin-Dynein Regulatory Complex (N-DRC) function in keeping the axoneme together during ciliary beating[7,8]. However, protofilament B9 is in an area of the microtubule that is inaccessible for IFT trains, thus unlikely to contribute to IFT regulation. Accordingly, depletion of glutamylation did not affect IFT motility[8]. Tubulin glycylation was instead found on subsets of protofilaments on both A-tubule and B-tubule that can be used by IFT trains[7]. Thus, while glycylation has a role in regulating axonemal dynein activity[9], and might affect IFT motors in general, it is unlikely to contribute to the mechanism of selection of either A-tubule or B-tubule by the trains. Tubulin detyrosination, on the other hand, was shown to be strongly enriched on the B-tubule of Chlamydomonas cilia by immunogold-EM[10] and sequential purification of tubulin from microtubule doublets[5], but its role in cilia remains elusive.

In mammals, it was recently discovered that the genetically encoded C-terminal tyrosine of alpha-tubulin is cleaved post-translationally after microtubule polymerization by tubulin carboxypeptidases such as the Vasohibin-SVBP complex[11] or MATCAP[12]. The Chlamydomonas homolog of these enzymes is not annotated, but detyrosination of axonemal microtubules is a conserved feature across ciliated species. In vitro studies have shown that mammalian kinesin-1[13–15] and kinesin-2[16] exhibit higher landing affinity and processivity on detyrosinated microtubules. Additionally, cytoplasmic dynein complexes have preferential affinity for tyrosinated microtubules via the CAP-Gly domain of its adaptor dynactin microtubule binding domain[17–19]. Thus, the observed differential distribution of tubulin tyrosination and detyrosination could encode signals on the A-tubule and B-tubule that might be read by IFT motors and contribute to the spatial segregation of retrograde and anterograde IFT trains, respectively.

In this study, we show that the absence of tubulin detyrosination in the cilia of Chlamydomonas mutants resulted in recurrent stoppages of IFT trains. Concurrent with these stoppages and alteration of IFT dynamics, we observed a reduction in the ciliary growth rate. The absence of structural defects in the mutant cilia, as revealed by cryo-electron tomography (cryo-ET), suggested a direct interaction between the IFT trains and tubulin detyrosination. To test this hypothesis, we developed a method to reconstitute the motility of native IFT trains on de-membranated cilia (parent axonemes) and ex vivo on synthetically polymerized microtubules enriched for different post-translational modifications. We found that anterograde and retrograde IFT trains have differential affinities for detyrosinated and tyrosinated tubulin. Based on our results we conclude that the enrichment of detyrosinated tubulin on the B-tubule[5,6,10] biases the sorting of IFT trains, by selectively recruiting anterograde trains while excluding retrograde trains. In the mutant cells, the absence of tubulin detyrosination removes this bias and induces the missorting of the IFT trains onto the wrong microtubules. This gives rise to disruptions in IFT dynamics and seamless transport of cargo within cilia, likely cause by train collisions.

## Results
### Depletion of tubulin detyrosination in Chlamydomonas *VashL* mutant cells leads to IFT train stoppages and slows down ciliary growth
To investigate the role of tubulin detyrosination in regulating IFT train motility in Chlamydomonas, we targeted the Cre05.g241751_4532

locus in the Chlamydomonas genome, which corresponds to the SVBP (NCBI RefSeq: NP_955374.1) peptide of the human Vasohibin-SVBP complex (see Materials and Methods) (Fig. S1). Screening of the Chlamydomonas Library Project (CLiP) revealed a mutant strain (LMJ.RY0402.233724) with insertion in the target locus. Additionally, to specifically disrupt the target locus we utilized CRISPR technology[20] to generate a mutant with impaired tubulin detyrosination activity (further referred to as *VashL*). Both CLiP and CRISPR mutants exhibited a strong reduction in the levels of detyrosinated tubulin (Fig. S1A). Further, the CRISPR mutant did not show any changes in other major tubulin PTMs (including Acetylation, polyglycylation and polyglutamylation), (Fig. S2B), suggesting that knocking out Detyrosination does not affect other major tubulin PTMs in the cilia. We performed the knock-out on wild-type cells as well as on a background strain containing fluorescently tagged IFT46 to study the dynamics of IFT trains in the absence of tubulin detyrosination by TIRF microscopy. CRISPR knock-out mutant (*VashL*) was used for all further experiments.

IFT trains in the *VashL* cells showed altered dynamics compared to wild-type cells. While trains moved predominantly in an uninterrupted manner in wild-type cells, they exhibited a recurrent 'stop-and-go' movement, that is, fast runs were frequently interrupted by transient stationary phases in the *VashL* cells (Fig. 1A, green or magenta arrowheads, see also Fig. S3). All observed stoppage events were associated with the crossing of opposite-polarity trains (white arrowheads), and no stoppages were detected in the absence of such crossing events. However, not every crossing event resulted in a train stoppage. Since most crossing events involved trains moving in opposite directions on separate microtubule doublets, the occurrence of stoppages during crossings suggests that anterograde and retrograde trains in VashL cells often collide when sharing the same doublet. These collisions likely cause the observed temporary train stoppages (see also Discussion). Quantification of individual stoppage events and normalization relative to total crossing events of each train revealed that anterograde trains were more likely to stop than retrograde trains after a crossing (Fig. 1B). This effect was independent of the total number of trains per cilium, as the train injection rates were similar in *VashL* and wild-type cells (Fig. 1C).

To further test if the observed alteration of IFT dynamics also leads to perturbations of cilia assembly rate, we measured the rate of cilia regrowth after deciliation in wild-type and *VashL* cells. While the steady state cilia length was not affected (Fig. S2C), the initiation of cilia regrowth in *VashL* cells was indeed delayed by about 15 min and progressed at slower rates than in wild-type cells, with mutant cilia never reaching the full length within the experimental time frame (Figs. 1D and S2C). Thus, depletion of tubulin detyrosination from axonemal microtubules induces recurrent IFT train stoppages and a reduced growth of cilia.

### Depletion of detyrosinated tubulin does not affect the axonemal structure
To examine the potential impact of detyrosination depletion on the axoneme structure, we utilized cryo-electron tomography and subtomogram averaging to compare wild-type and *VashL* axonemes. The comparison of the 3D structures of the microtubule doublet 96 nm repeat, up to a resolution of 30 Å, revealed no architectural changes or defects in A-tubules, B-tubules, and major axonemal components (Fig. S4). These results show that compromised tubulin detyrosination does not disrupt the assembly of a proper axonemal structure assembly. Thus, the observed cilia growth and IFT motility defects likely stem from a direct effect of the depletion of tubulin modification on the IFT trains and their motors, rather than microtubular or axonemal structural anomalies.

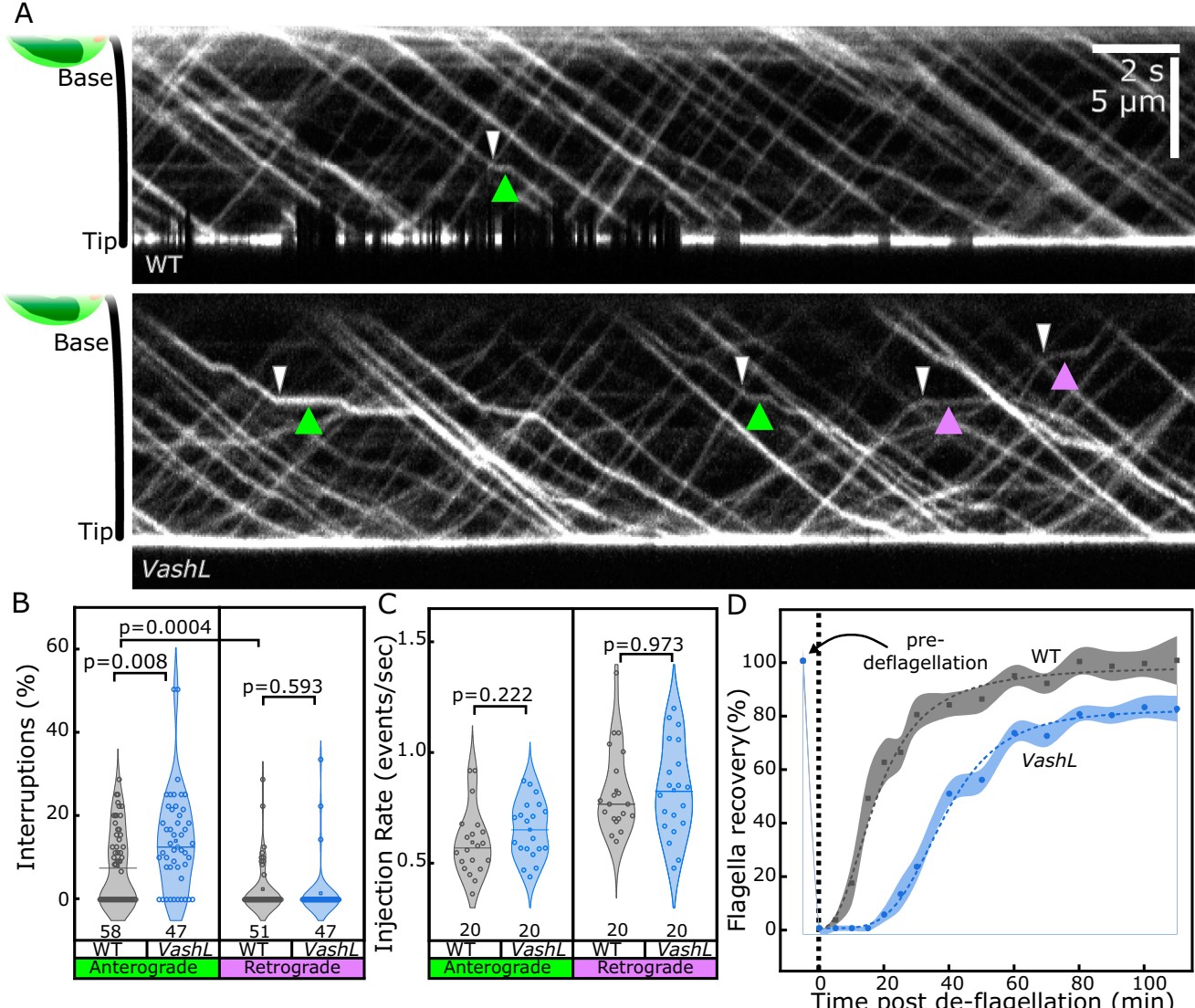

**Fig. 1 | IFT motility in VashL cells is frequently interrupted by stoppages.**
**A** Kymographs of wild-type (WT, top) or VashL (bottom) IFT trains. In *VashL* cells, anterograde or retrograde train stoppages (green or magenta arrowheads respectively) often occur after crossing events (white arrowheads, see Fig. S3 for additional examples). **B** Violin plots of interuptions during runs in wild-type (grey plots) or VashL (blue plots) trains. Each datapoint represents an individual train. Horizontal line represents median of each dataset. Total number of trains analyzed in each case as shown. Pooled data from 20-30 flagella, 2-3 independent experiments. Statistics by Two-sided Wilcoxon Signed rank test. *p*-values as shown.

**C** Violin plots of train injection rates from base or tip in wild-type (grey plots) or *VashL* (blue plots) cilia, respectively. Horizontal line represents median of each dataset. Total flagella analysed in each case as shown, 2–3 independent acquisitions. Statistics by Student's two-tailed t-test. *p*-values as shown. **D** Kinetics of cilia recovery in wild-type (grey plot) or *VashL* (blue plot) cells, expressed as % w.r.t. cilia length pre-deciliation. Error bands, datapoints and dashed lines represent 95% CI, measured averages and logistical curve fits respectively. N = 12–35 cells for each time point, pooled data from 2 independent replicates for each case.

## Reconstitution of IFT train motility ex vivo on synthetically polymerized microtubules

To test whether the train stoppage events observed in vivo are caused by a direct effect of the tubulin detyrosination depletion, we developed an ex vivo approach where we reconstitute the motility of detergent-extracted IFT trains from cilia of *Chlamydomonas* cells on in vitro polymerized, polarity-marked microtubules (Fig. 2, Materials and Methods). Specifically, in vitro polymerized microtubules were applied onto a coverslip and imaged on a TIRF microscope. *Chlamydomonas* cells expressing fluorescently tagged IFT proteins were overlaid onto the microtubules in a droplet of ATP-containing buffer, which supported IFT motility (Fig. 2). A capillary micropipette filled with membrane solubilizing agent (1% Igepal CA-630, henceforth 'detergent') was immersed in the droplet and centred over the field

of view using a stage-mounted micromanipulator (Fig. S6A). A micro-dose of detergent was delivered over the *Chlamydomonas* cells using a calibrated microinjector pump to gently demembranate the cilia and efficiently extract IFT trains (Fig. S5; Fig. S6D, S6E, Supplementary Movie 2). The detergent shot from the capillary pipette caused rapid demembranation of the cilia but did not affect the membrane of the cell body, as it is protected by the algae cell wall. Further, as the effective bulk dilution of the detergent was more than 1:10^9 (20–40 fL (Fig. S5) in 5–10 µL of droplet) cilia of cells outside the area of effect were not demembranated. This enabled repeated detergent deliveries in multiple areas within the same droplet, improving throughput and data acquisition. Once extracted from the cilium, IFT trains landed and processively moved along the in vitro polymerized microtubules.

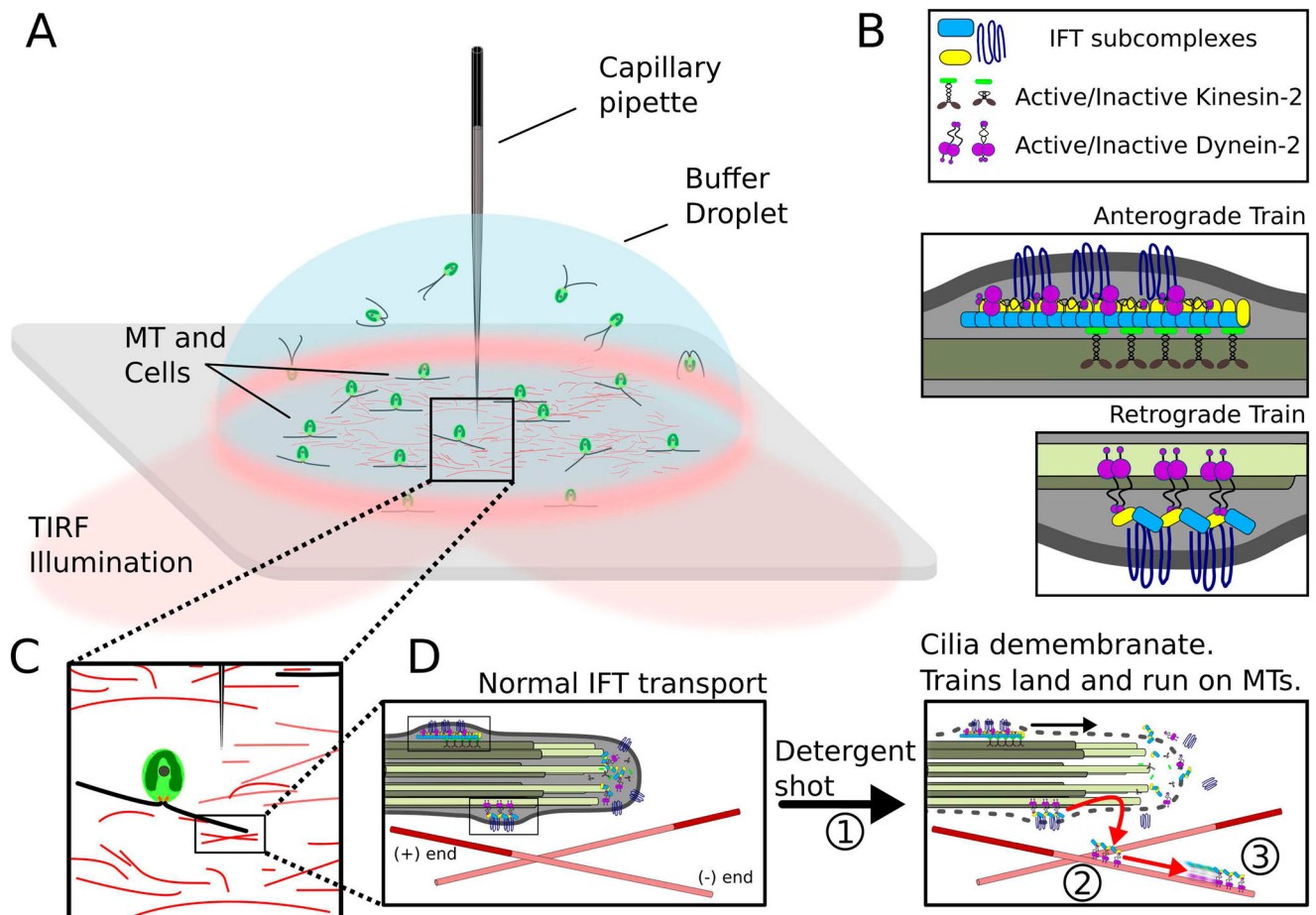

**Fig. 2 | TIRF microscopy coupled with targeted cilia demembranation to reconstitute IFT train motility ex vivo. A** Illustration of the reconstitution setup. Chlamydomonas cells are overlaid on synthetically polymerized microtubules adhered to a coverslip in a droplet of motility buffer on a TIRF microscope. A detergent back-filled capillary micropipette is immersed in the buffer from the top and centred using a three-axis micromanipulator. **B** Legend showing components of IFT trains (Top), and illustrative representations of an anterograde (Middle) and retrograde train (Bottom). **C** Illustration of relative positions of microtubules, cells, and capillary pipette. **D** Illustration of in vivo IFT train motility (Left) and ex vivo IFT train motility on polarity-marked microtubules (Right). ① Detergent shot from capillary pipette solubilizes ciliary membrane. ② IFT trains are released into the surrounding environment. Some motile trains continue to move (black arrow) on the demembranated cilia (parent axoneme), while ③ others land and move (red arrows) on the polarity-marked microtubules (see also Supplementary Movie 3).

Typically, the extracted trains (fluorescently labelled IFT-B, IFT46-mNeonGreen) landed on a microtubule, resumed motility, halted after traveling few micrometres (2.7 ± 2.1 µm, mean ± SD, *n* = 633 trains) and did not resume motility thereafter (Fig. 3A, S7, Supplementary Movies 3 and 4). In rare cases, IFT trains detached from the microtubules instead of halting (Fig. S7). We hypothesized that train halting ex vivo could be caused by microtubule obstacles, such as other ciliary components interacting with the microtubules, or the loss of motor activity outside the ciliary environment. To test for presence of obstacles, we demembranated cilia of different strains expressing fluorescently labelled non-IFT ciliary components FMG1-B[20], RSP3[21] or ODA6[22]. No landing or motility events on microtubules were detectable in these cases (Supplementary Movies 5–7). This showed that we can specifically reconstitute the motility of IFT trains on microtubules ex vivo and that non-IFT components from demembranated cilia did not interfere with motility of ex vivo IFT trains in our assay.

### All main constitutive sub-complexes of IFT trains are retained ex vivo

To verify whether the extracted IFT trains retained their overall in vivo composition, we repeated the reconstitution experiment with strains containing fluorescently labelled IFT-A (IFT140-sfGFP) and IFT-Dynein

(d1bLIC-GFP). For all strains, IFT trains were consistently landing and moving on microtubules, demonstrating that all IFT complexes (IFT-A, IFT-B and the motors) are retained in the trains ex vivo (Fig. S8, Supplementary Movies 8 and 9). The average ex vivo velocities of anterograde and retrograde IFT trains were consistent between each strain (Fig. 3B) and comparable to the velocities observed in vivo (Fig. 3C). Additionally, previously reported single-molecule velocities for the *Chlamydomonas* kinesin-2 complex[23] were comparable to ex vivo anterograde train velocities. We conclude that ex vivo reconstitution could recapitulate in vivo IFT train behaviour. Further as the tested IFT-labelled candidates are known to be stably associated with their respective sub-complexes after extraction[24–26], we concluded that reconstitution could recapitulate overall train sub-complex composition.

### Anterograde trains have stronger affinity for parent axonemes than retrograde trains and the absence of tubulin detyrosination reduces their affinity for parent axonemes

When quantifying the relative amount of anterograde and retrograde IFT trains on the synthetically polymerized microtubules, we found that a vast majority of ex vivo IFT trains moved in retrograde direction consistently between different cell types (Fig. 3D). This suggested that

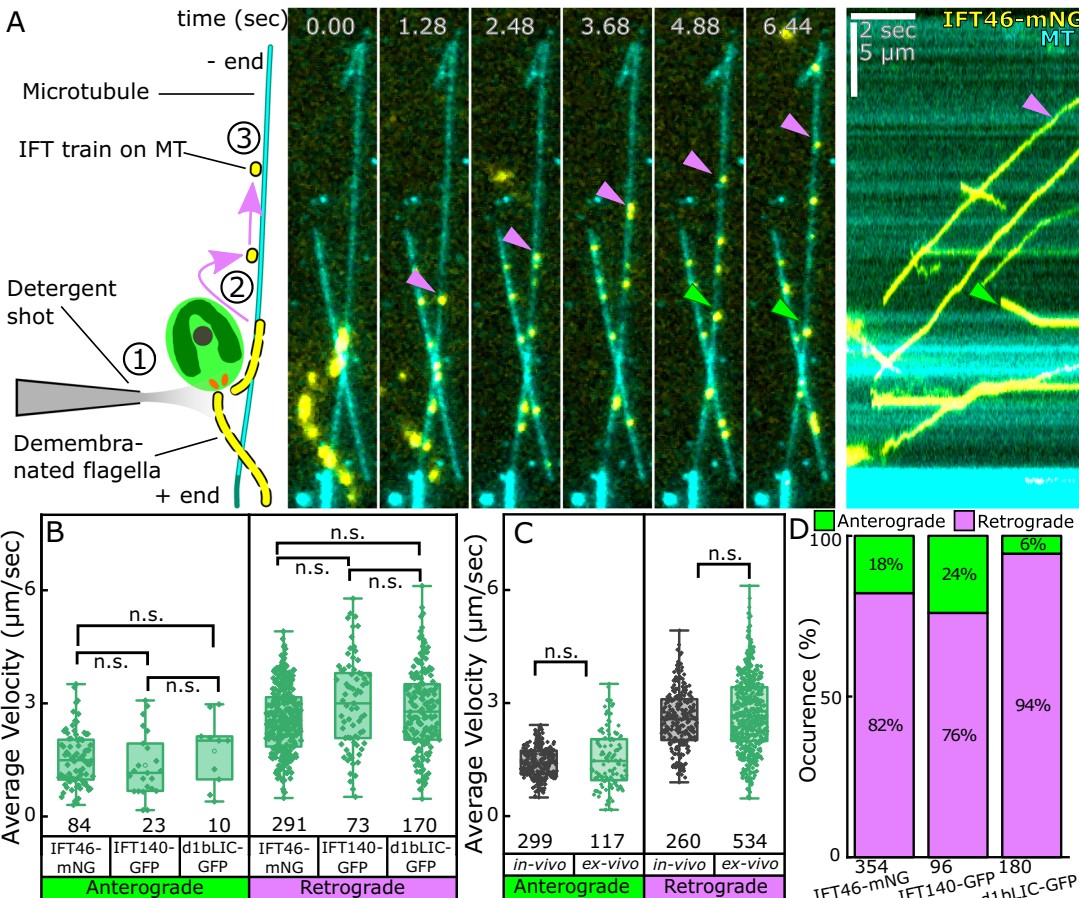

**Fig. 3 | Velocities of IFT trains are conserved during ex vivo reconstitution.**
**A** Left: Schematic of ① detergent shot, ② landing of IFT train and ③ motion on microtubule. Right: Representative montage and kymograph of reconstitution of IFT trains (magenta and green arrowheads) comprising mNeonGreen labelled IFT46. The directionality of train movement is determined by plus-end labelling of microtubules (see Materials and Methods, Supplementary Movie 4). **B** Box plots of average velocities as measured from kymographs in (**A**), for ex vivo IFT trains with fluorescently labelled IFT46 (IFT-B), IFT140 (IFT-A), or d1bLIC (IFT Dynein) (see also Fig. S7). Each point represents individual train analyzed as shown, from 50–70 cells across 6-7 biological replicates in each case. Box bounds represent 25% and 75% quartiles. Line represents median. Box whiskers represent maxima and minima C) Box plots of average in vivo (black) or ex vivo (green) velocities pooled from (**B**). For both (**B**) and (**C**), statistics by Student's two-tailed t-test. n.s., $p$ = not significant. Each point represents individual train analyzed as shown, from 50–70 cells across 6-7 biological replicates. Box bounds represent 25% and 75% quartiles. Line represents median. Box whiskers represent maxima and minima (**D**) Stacked bar plots of directionality distribution of ex vivo trains (labelled labelled by either IFT46-mNeonGreen, IFT140-sfGFP or d1bLIC-GFP). Total trains analyzed as shown from 100–200 demembranation events and 2-3 independent replicates, similar to B.

anterograde trains were either less likely to be extracted from cilia or would land less frequently on the microtubules.

To test whether the efficiency of IFT train extraction from cilia was comparable between anterograde and retrograde trains, we demembranated cilia of IFT46-mNeonGreen cells and estimated the number of anterograde or retrograde trains that continued to move along the demembranated cilia (henceforth "parent axoneme") as fraction of the total number of respective trains in the flagella shaft before demembranation (Fig. 4, Supplementary Movies 10 and 11). We observed that anterograde trains were unlikely to detach from wild-type parent axonemes. Only a small fraction of them left the axoneme and landed on the synthetically polymerized microtubules. Contrary to this, retrograde trains had a high detachment probability and often landed on the microtubules (Fig. 4). These observations indicate that anterograde trains have a stronger affinity for the parent axoneme than retrograde trains.

Interestingly, when we exposed the detyrosination deficient *VashL* cells to the same detergent treatment, most anterograde trains quickly left the parent axoneme with an increased detachment probability, while the detachment probability was reduced for retrograde trains (Fig. 4A, B). In view of these results and because of the recorded

high rate of IFT train stoppages in the *VashL* mutant, we propose that depletion of detyrosinated tubulin reduces the affinity of the anterograde IFT trains for the (normally detyrosinated) B-tubules. We suggest that the absence of tubulin detyrosination in *VashL* axonemes interferes negatively with anterograde IFT train affinity and positively with retrograde IFT train affinity.

## Anterograde trains are more likely to land on detyrosinated microtubules and retrograde trains on tyrosinated microtubules

To assess the ability of tyrosinated and detyrosinated microtubules to recruit oppositely directed IFT trains, we estimated the relative train landing probabilities on microtubules with varying amounts of tyrosinated tubulin. To achieve enrichment or depletion of tyrosinated tubulin, brain tubulin was treated with either tubulin tyrosine ligase[27] or carboxypeptidase-A, respectively (Material and Methods). Untreated microtubules were used as controls. To estimate the IFT train landing probability, motile ex vivo IFT trains were counted per unit length of microtubule per demembranation event. The landing probability of anterograde trains was found to be higher on microtubules treated by carboxypeptidase-A as compared to untreated

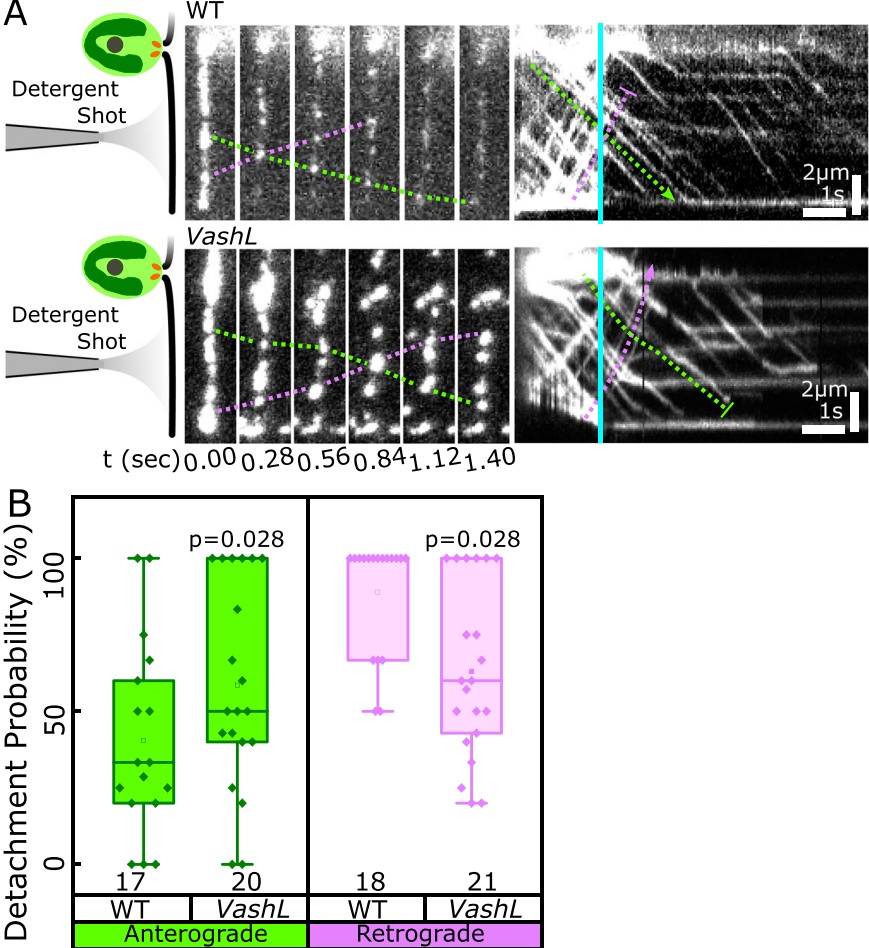

**Fig. 4 | Detachment kinetics of IFT trains from parent axonemes of wild-type or *VashL* cells. A** Schematics, representative montage and kymograph of demembranated wild-type (WT) and *VashL* parent axonemes (vertical cyan lines indicate time point of demembranation). In wild-type cells, anterograde IFT trains were less likely to detach from parent axonemes after demembranation (green dashed connectors) as compared to the retrograde IFT trains (magenta dashed connectors). In *VashL* cells, anterograde IFT trains were more likely to detach from the parent axoneme than in wild-type, and retrograde trains are retained more than anterograde trains. **B** Box plots of detachment probabilities of anterograde (green) and retrograde (magenta) IFT trains in wild-type and *VashL* cells. Statistics by Two-sided Wilcoxon Signed Rank Test. *p*-values as shown. Each point represents individual axoneme analyzed as shown, from 3 biological replicates each. Box bounds represent 25% and 75% quartiles. Line represents median. Box whiskers represent maxima and minima.

microtubules. In contrast, retrograde trains had a higher landing probability on microtubules treated by tyrosine ligase (Fig. 5A, B). To underscore the preference of anterograde trains for detyrosinated tubulin, and retrograde trains for tyrosinated tubulin, we estimated the relative IFT train landing probabilities with respect to the relative normalized tubulin tyrosination in each case (Fig. 5C, D, see also Fig. S9A, S9B). We find that the landing probability of anterograde IFT trains decreases (Fig. 5C), while the landing probability of retrograde IFT train increases (Fig. 5D) as function of increasing tubulin tyrosination. Taken together, we suggest that anterograde trains preferentially land on detyrosinated microtubules, while retrograde trains preferentially land on tyrosinated microtubules.

### Tubulin tyrosination/detyrosination does not affect train velocities

Finally, we measured the velocities of anterograde and retrograde trains on untreated microtubules as well as on microtubules treated with carboxypeptidase-A or tubulin tyrosine ligase. Unlike the landing probabilities, the average train velocities were unaffected on tyrosinated or detyrosinated microtubules (Fig. S9C). This further supports the hypothesis that the slowdown of trains observed

in *VashL* cells (Fig. S2B) is not directly due to a lack of detyrosination, but rather due to the increased number of stoppage events (Fig. 1A, B).

## Discussion

In this work we used CRISPR-mediated targeted mutagenesis to show an effect of depletion of tubulin detyrosination in impairing IFT motility and logistics. Loss of detyrosination led to recurrent temporary stalling of IFT trains and further a discernible slowdown of cilia elongation/regeneration but did not affect the structural integrity of the axoneme. A parallel can be drawn to the phenomena observed in the context of glutamylation and glycylation in *Chlamydomonas* and mouse axonemes, wherein the absence of these specific post-translational modifications does not affect the structure and, despite causing motility phenotypes, does not completely obliterate the beating of the axoneme[7–9,28]. We conclude that tubulin post-translational modifications do not orchestrate the process of axonemal structure assembly, but rather the fine regulation of axonemal and IFT constituents.

With the development of our method for the reconstitution of the motility of IFT trains ex vivo, we demonstrated that tubulin detyrosination and tyrosination have a direct effect on the affinity of

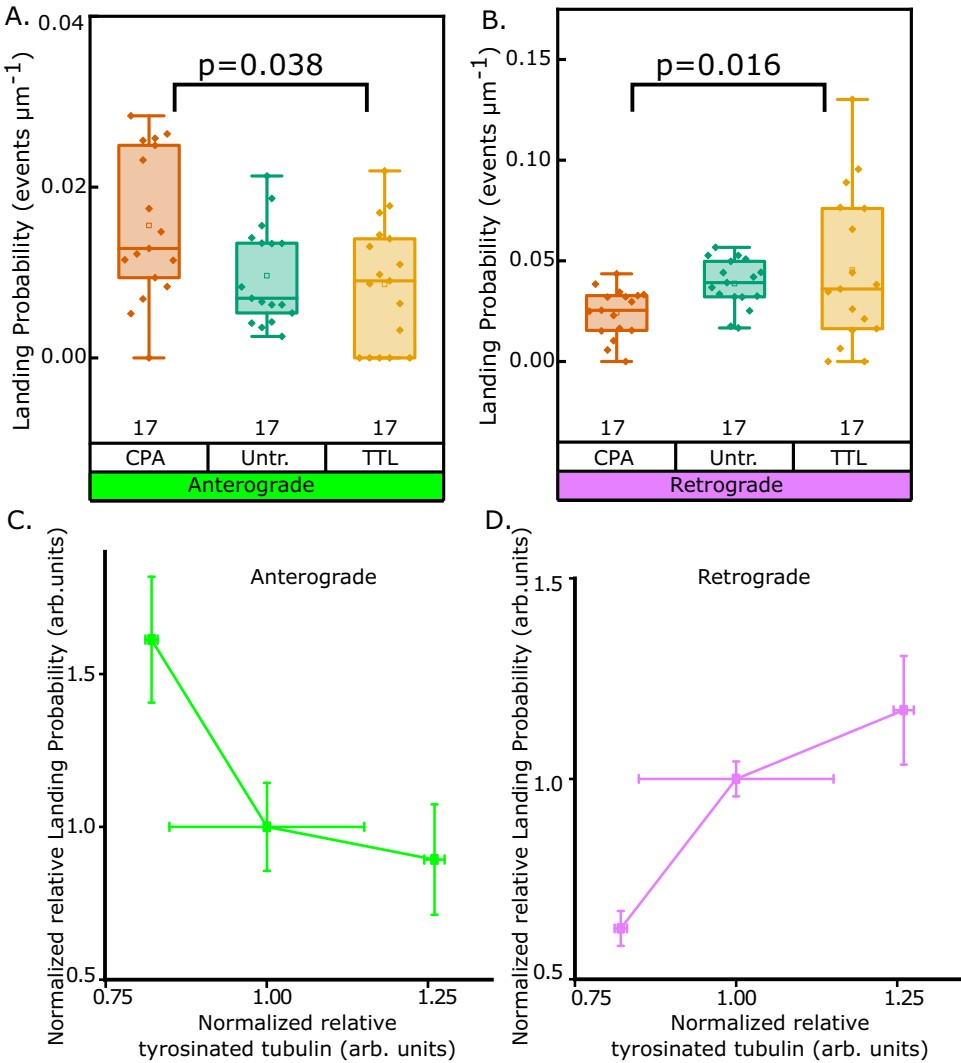

**Fig. 5 | Landing probabilities of ex vivo IFT trains on detyrosinated/tyrosinated microtubules. A**, **B** Box plots of landing probability of anterograde or retrograde IFT trains on untreated (green plots), carboxypeptidase-A treated (orange plots), or tubulin tyrosine ligase treated (yellow plots) microtubules. Each data point indicates individual demembranation events in each case, across 3 biological replicates in each case. Statistics by one-way ANOVA. *p*-values as shown. Box bounds represent 25% and 75% quartiles. Line represents median. Box whiskers represent maxima and minima. **C**, **D** Relative change in anterograde (green plot) or retrograde (magenta plot) IFT train landing probabilities (normalized within samples from (**A**) and (**B**)) w.r.t. relative microtubule tyrosination (see Fig. S9). n and N as in (**A**) and (**B**) for each case. Erros bars represent Mean ± S.E. on both axes.

anterograde and retrograde trains to the respective microtubules. Anterograde trains showed higher affinity for detyrosinated axonemes in wild-type cells and preferential landing on reconstituted detyrosinated microtubules. Retrograde trains displayed stronger affinity for the tyrosinated axonemes of *VashL* mutant cells and preferential landing on reconstituted tyrosinated microtubules.

Our in vivo observations of recurrent stoppage of IFT trains in the *VashL* detyrosination-deficient mutant, leads us to conclude that the differential affinity for detyrosinated and tyrosinated microtubules by IFT kinesin and dynein motors as seen in vitro has a specific role in the regulation of IFT logistics in cells. We reason that the mutant phenotype can be attributed to a uniform distribution of tyrosination over both A-tubule and B-tubule of the microtubule doublets, unlike the normal phenotype, where B tubule is detyrosinated and A tubule is tyrosinated[5,6,10]. The consequent loss of specificity for the B-tubule and A-tubule by opposite polarity anterograde and retrograde trains may cause the trains to travel on any microtubule and eventually end up in collisions, likely leading to the observed stoppage events (Fig. 6B). Theoretically, the chance that a retrograde train physically interacts with a given anterograde train at a crossing relies on it being on the same of the two tubules of a given doublet. The overall probability of a collision is ~6% (= 1/2 ×1/9 = 0.5 ×11.11%). On the other hand, stoppages due to collisions are unlikely in wild-type cells (as opposite polarity trains do not walk on the same tubule of any microtubule doublet[3]). Consistent with this expectation, we observe anterograde trains stopped ~8% more often in *VashL* cells than in wild-type cells (Fig. 1B). Additionally, the finding that we have not observed a 'stop-and-go' behaviour for trains ex vivo especially on tyrosinated microtubules, strengthens our hypothesis that the observed increase in the stoppage events in the *VashL* cells points towards the presence of collisions. In wild-type cilia, despite close physical proximity of both trains to either tubule, asymmetric fidelity of train sorting is likely maintained by the specific distribution of tyrosinated and detyrosinated tubulin and head-on collisions with oncoming trains are avoided.

Based on our experimental findings, we suggest a role for tubulin tyrosination/detyrosination in regulating the dynamics of IFT trains for efficient transport in cilia. We propose a model where train localization is encoded directly on the microtubule tracks. This does not exclude additional regulatory mechanism of microtubule selection by the oppositely directed trains. It has been shown that assembling

**A Wild Type**

Detyrosinated B-Tubule

Tyrosinated A-Tubule

**B *VashL* Mutant**

① ② ③

Tyrosinated A and B-Tubule

**Fig. 6 | Proposed model for the mechanism of IFT train sorting governed by tubulin tyrosination/detyrosination. A** In wild-type cells, anterograde IFT trains get loaded and are retained on the B-tubule (majorly detyrosinated, dark olive). Proximal to the ciliary tip, retrograde IFT trains are conversely loaded and retained on the A-tubule (majorly tyrosinated, light olive), thereby ensuring that train collisions are avoided. **B** In *VashL* mutant cells, ① anterograde IFT trains do also get loaded onto the B-tubule (tyrosinated, light olive). ② However, due to an increased detachment probability, anterograde IFT trains deviate onto the A-tubule (tyrosinated, light olive). ③ Oncoming retrograde IFT trains then collides with missorted anterograde IFT trains leading to stoppage events. An analogous missorting of retrograde trains onto B-tubules is also possible.

anterograde trains at the ciliary base associate with the B-tubule already at the transition zone[29]. Thus, in wild-type cells, detyrosination of B-tubule[5,6,10] might play a critical role in positively selecting anterograde trains at the ciliary base and then retaining them on the B-tubule along the shaft. A similar mechanism could exist in IFT models that use two-Kinesin systems for anterograde transport, particularly since both heterotrimeric and homodimeric IFT kinesins belong to the same kinesin family. It would be interesting to study of the role of tubulin PTMs in anterograde cargo hand-off in future. At the Chlamydomonas ciliary tip, the A-tubules extend further than the B-tubules[30]. Thus, retrograde trains might naturally land on A-tubules initially owing to their proximity. We suggest that retrograde trains are then restricted to the tyrosinated A-tubules because they are excluded from the B-tubules due to their lower affinity for detyrosinated tubulin (Fig. 6A).

In addition to the direct influence of tubulin post-translational modifications, other mechanisms could facilitate the avoidance of collisions and control tubule-specific transport of anterograde and retrograde IFT trains. For example, single molecule motility assays indicated an adaptive stepping behaviour of ciliary kinesin-2 wherein the intrinsic ability of the motor to switch protofilaments of microtubules was explained by a sidestepping model of IFT train sorting on the microtubule doublets[31]. While direct evidence for sidestepping of rigid linear arrays of multi-motor complexes like a IFT train on microtubule doublets is lacking, the possibility of motor sidestepping functioning cooperatively and redundantly alongside our proposed mechanism for efficient train sorting cannot be ruled out. However, we observed an increased number of stoppage events in *VashL* cells (Fig. 1B), where motor sidestepping is likely not supressed. This indicates that sidestepping alone might not be sufficient to avoid the collisions of oppositely directed IFT trains and that tubulin detyrosination is necessary to regulate IFT motility.

Vash-SVBP complex binds to the microtubule lattice via the N and C terminal disordered domains, and directly cleaves Tyrosine on lattice[32]. As there exist no known detyrosinaton sites on IFT complex proteins, we believe that affinity bias of trains on microtubules is driven by tyrosination/detyrosination of microtubule doublet lattice by these enzymes, rather than potential tyrosination/detyrosination of other IFT proteins.

The role of Vasohibin in tubulin detyrosination is described in other models, such as lymphatic angiogenesis in zebrafish[33], or regulation of cell cycle and morphogenesis in Trypanosoma parasites[34]. Here we have reported its role in cilia. Similar regulation could exist in higher organisms, such as in sperm, although concrete evidence for this is missing. While other tubulin PTMs like polyglycylation indeed influence sperm motility[9], SVBP-KO mice are fertile but have cognitive abnormalities[35], suggesting that the tubulin Tyrosination/Detyrosination may have organ and tissue specific function.

Analogous to proposed sub-cellular localization of tubulin PTMs in cilia, local differences in polarity or PTM state of microtubule bundles biases local transport and organelle sorting as a function of microtubule stability and age[36]. In vitro studies have shown that mammalian kinesin-1[14] and kinesin-2[16] exhibit increased landing affinity and processivity for detyrosinated microtubules, although such affinity bias has so far not been reported for *Chlamydomonas* IFT kinesin. Additionally, cytoplasmic dynein complexes have shown preferential affinity for tyrosinated microtubules via the CAP-Gly domains of their adaptor dynactin microtubule binding domains[37]. However, we and others have previously not found any indication of a CAP-Gly domain-like sensor within any component of the IFT complex[2,24]. Apart from the IFT motors themselves, IFT81/74[38] and IFT54[39] have tubulin binding domains. IFT81/74 is relatively far from the microtubule doublet on anterograde trains[2], and rather binds to soluble α/β tubulin dimer as a cargo adapter[38]. It is therefore unlikely that IFT81/74 reads the detyrosination status of doublet tubules. Also, IFT54 binds directly to both IFT kinesin and dynein[40] and reads only the E-hooks of tubulin or microtubules via its CH-domain[39]. It remains unknown how IFT senses tubulin tyrosination/detyrosination at a molecular level or how this or other PTMs are finely deposited with sub-cellular precision.

Methodologically we foresee a wide range of applications for our ex vivo/in vitro approach. While in vitro assembly and motility of IFT protein complexes from constitutive proteins has been achieved[41], reconstituting full IFT trains bottoms-up is a formidable challenge. Likewise, although IFT subcomplexes have been isolated using traditional protein isolation methods[42–44] biochemical purification of entire IFT trains has been unsuccessful so far, mostly due to the complex multimeric nature and distinct structures for anterograde and retrograde IFT trains. IFT trains were seen running only for short periods

along the axonemal microtubules of permeabilized primary cilia[45]. Hence, our method, together with the 'molecular motor toolbox'[46], presents an opportunity to explore the behavior of native trains in a highly controllable environment to study how other factors, such as viscosity, temperature, and the presence of microtubule-associated proteins[47] and microtubule internal proteins will differentially influence the motility of anterograde and retrograde IFT trains.

## Methods

### Chlamydomonas cell culture
*d1bLIC::D1bLIC-GFP* (CC-4488), *oda6::IC2-NG*(CC-5857), LMJ.RY0402. 233724 were purchased from Chlamydomonas Resource Center. *FMG1B*-mNeonGreen (CC-6012) and *ift46::NIT IFT46-mNeonGreen* (+) (CC-5900) was grown from lab stock. *ift140-1::NIT1 IFT140-sfGFP, pf14::RSP3-NG* were kind gifts from Karl Lechtreck, University of Georgia.

Cells were inoculated from TAP-agar plated into liquid TAP medium grown with continuous bubbling of sterile air and normal 12 h:12 h light-dark cycle.

For de-membranation experiments, ~5–10 mL cells were harvested, and washed thrice with TAP containing 1 mM EGTA. For final mounting, a small drop (~2–4 µL) of double-stabilized microtubule suspension was overlaid with cell suspension (final cell density ~10⁶ /mL), in neutral TAP, with 10 µM paclitaxel and 1 mM ATP final.

### Creation of VashL IFT46-mNeonGreen
*ift46::NIT IFT46-mNeonGreen* (+) is a back-cross of CC-5900 and CC-125. The Chlamydomonas Vash2-SVBP homologue *VashL* (Cre05. g241751_4532) was disrupted by CRISPR/Cas9 insertional mutagenesis with a cassette targeted to the CRISPR guide ATGTGATACCGGC ACCACTG**TGG** on the first exon about one kilobase downstream of the start codon. The cassette is composed of 50 bp homology arms up- and downstream of the cut side flanking a nourseothricin acetyl transferase (NAT) gene under the control of a RbcS2 promoter-terminator pair. Transformation of the *ift46*::NIT IFT46-mNeonGreen (+) was performed as previously described[48]. In brief, cell walls were removed with three incubations in gamete autolysin over 2 hours. During the final round of autolysin incubation, cells were heat shocked at 40 °C for 30 minutes, followed by three washes in TAPS medium (TAP + 40 mM sucrose). 10⁶ cells in 5 µM were mixed with 5 µL RNPs at 5 µM and 1500 µL of BspQI digested and purified donor plasmid. The transformation mixture was electroporated in a 10 µL Neon electro-porator at 2300 V, 12 ms, 3 pulses and ejected into 1 mL TAPS for overnight recovery. Finally, cells were selected on 1.5% TAP-agar plates supplemented with 7.5 µg/mL nourseothricin. Resulting colonies were picked into a 96-well plate filled with TAP and screened for a correct insertion by PCR and sequencing using primers flanking the cut site.

### Coverslip preparation
22 mm * 22 mm Glass coverslips (Menzel-Glaser, #1.5) were cleaned by sonication in Mucasol/water (1:20; v/v) for 15 min followed by rinsing in deionized water for 2 min. Further, coverslips were soaked in 0.05% solution of Dichlorodimethylsilane (Sigma) for 60 min, rinsed and sonicated twice with 100% methanol, and twice with Milli-Q for 10 min each, and blow dried with nitrogen gas.

### Capillary pipette preparation and manipulation
Capillary pipettes were prepared from thin borosilicate capillaries (BF100-50-15, Sutter Instruments) using micropipette puller (P-1000, Sutter Instruments) as per standard manufacturer protocol. Pipette orifice diameter was maintained at about 2 µm. Taper length of neck was 6-8 mm to ensure sufficient pipette flexibility. Prepared capillary pipettes were back-filled with 1% Igepal CA630 solution (~5 µL) and manipulated with a combination of coarse (MN-4, Narishige, Japan) and fine (MMO-4, Narishige, Japan) manipulators. Solution ejection

parameters (ejection pressure, time pulse) were controlled by a micropipette pump (PV-850, World Precision Instruments, UK).

### Pipette centring
Vertically mounted capillary pipette was carefully centred and focused to the field of view while imaging live through 10X, 40X, and finally 100X objective using the drive unit of micromanipulator. The working position of the pipette was ~10–20 µm above the glass surface. Ejection volume calibration was performed according to capillary manufacturer protocols, by ejecting solution into paraffin oil and estimating the volume of the sphere thus obtained. Ejection parameters were ramped as per manufacturer's recommendation to tune optimal desired ejection volumes. The area of effect for a capillary with orifice diameter of approximately 2 µm was limited to a circle with a radius of roughly 15 µm (Fig. S6B, S6C, see also Supplementary Movie 1).

### Microtubule polymerization and polarity labelling
Tubulin was purified from porcine brain (Vorwerk Podemus, Dresden, Germany) using established protocols as described previously[49].

Polarity-marked microtubules were prepared by preferably extending the plus-ends of long GMPCPP seeds in the presence of N-ethylmaleimide (Sigma) modified tubulin (NEM-tubulin). Briefly, long dim seeds were polymerized from 1:9 rhodamine labelled tubulin (final conc. 2 µM) in the presence of 1 mM GMPCPP (NU-402, Jena Bioscience) in BRB80 at 37°C. For plus end labelling, an extension mix comprising 4 µM 1:3 rhodamine labelled tubulin, 1 µM fresh NEM-tubulin (40 µM unlabelled tubulin incubated with 0.4 mM NEM on ice for 10 min and excess NEM quenched with 20 mM DTT for 10 min), 1 mM GTP(Jena Biosciences) and 4 mM MgCl2 in BRB80 was assembled on ice, warmed at 37 °C for 1 min and incubated with 1/10th volume of dim seeds at 37 °C for 60 min. Microtubules were stabilized with 10 µM taxol in BRB80 (Paclitaxel, Sigma) and harvested by spinning at 17,000 g for 15 min. Polarity-marked microtubules were used on the same day of preparation.

### Enzyme treatment of microtubules
For tyrosination, a previously described protocol[27] was adopted with some modifications. 40 µM 1:9 rhodamine labelled tubulin (final conc. 13.33 µM) was incubated with 5 µM Tubulin-Tyrosine Ligase (from Michel Steinmetz, see Acknowledgements), in presence of 0.1 mM sodium salt of L-Tyrosine (Merck), 150 mM KCl, 12.5 mM MgCl₂, 1 mM DTT and 2.5 mM ATP in BRB80 buffer. The reaction was incubated at 25°C for 3 hours and stopped by adding 1 mM PMSF. For detyrosination, tubulin stocks were prepared as described before[37] with slight modifications. Briefly, 1:9 rhodamine labelled tubulin (final concentration 72 µM) was incubated with Carboxypeptidase-A (Merck) at a relative ratio of tubulin:enzyme (3:1 w/w) at 37 °C for 10 min, in presence of 20% (v/v) glycerol, 2 mM GTP. Reaction was stopped by adding 20 mM DTT. Treated tubulin was cycled once in 2 mM GTP at 37 °C to remove residual enzyme, flash frozen in liquid nitrogen and finally stored at -80 °C. For experiments, same batch of treated tubulin was split in equal parts and used for Western blot normalizations and ex vivo motility respectively.

### Microscopy
TIRF Imaging was done on a Ti2 Eclipse (Nikon, Japan) equipped with a 100x/1.49NA TIRF objective (MRD01991, Nikon, Japan) with a galvo-driven orbital TIRF consisting of a FRAP/TIRF combination system (iLas2, Gataca Systems, France) coupled to a multi-laser combiner (VS-LMS-MOT100, Visitron Systems, Germany). Multi-color, simultaneous acquisition was done by EMCCD cameras (iXon life, Oxford Instruments, UK) controlled by VisiView software 5.0 (Visitron Systems, Germany). Data was acquired as continuous stream at 40 ms frame time and 1024 * 1024 pixels at a resolution of 0.086 µm px⁻¹. Room and

sample temperature were maintained at 22 °C throughout the experiments.

For cell swimming, flow chambers were made by mounting a thin coverslip (0.17 mm) over a glass slide using double sided sticky tape (Tesa 05338). Liquid culture cells grown as explained above were pipetted inside the flow chamber right before imaging. An inverted Nikon Ti2 microscope was used in diascopic imaging mode using a white light transilluminator lamp (Nikon Intensilight C-HGFI). Movies were acquired at 387.4 fps and 0.564 um/pixel using a 20x objective (Plan Apo lambda, NA = 0.75, WD = 1000um) and a PRIM 95B 25MM camera (Teledyne Photometrics). All movies were acquired focusing in the middle of the flow chamber. The acquired movies were analyzed using the TrackMate plugin in Fiji. A custom Python script was used to compute swimming speeds, beat frequencies and forward and backward displacements per beat cycle.

### Analysis of TIRFM data
Each detergent ejection and de-membranation event were manually trimmed from a continuous movie stream in Fiji. Kymographs were reconstructed using spline ROIs (width = 3 px) over microtubules. Velocities were calculated from kymographs using custom written Fiji macros.

**Kymographs of Parent Axonemes.** Kymographs were traced over all possible locations of axonemes. The resultant kymographs were aligned using StackReg plug-in in ImageJ 1.54 f (http://bigwww.epfl.ch/thevenaz/stackreg/). Finally, the registered stacks were Z-projected for maximum intensity to visualize full coverage of axoneme temporally over the de-membranation period.

**Calculation of Detachment Probability.** For each train type on each parent axoneme, 'Retention fraction' for anterograde or retrograde trains was calculated as,

$$R_f = N_{pa}/N_c \tag{1}$$

where,

$N_{pa}$ = Number of trains per demembranated parent axoneme
$N_c$ = Number of trains per cilium before de-membranation
Newly injecting trains after de-membranation (particularly anterograde trains) were disregarded in the above evaluation. Further, 'Detachment Probability' was calculated as,

$$P_d = (1 - R_f) \times 100 \tag{2}$$

### Cilia isolation
Protocol for Isolation of *Chlamydomonas* cilia using pH shock method has been described in detail elsewhere[50]. Briefly, a bubbling culture of cells was washed in HMDEK buffer (30 mM HEPES, pH 7.4, 5 mM $MgSO_4$, 1 mM DTT, 0.5 mM EGTA, 25 mM KCl) thrice. Deciliation was done by adding 0.5 N acetic acid dropwise till pH of the buffer was reduced to 4.5 for 45 to 60 secs, before neutralization to pH 7.0 with 0.5 M KOH. Isolated cilia were suspended in HMDEK buffer containing 1 mM DTT, 1 µM Aprotinin and then were separated from cell bodies by spinning over a sucrose cushion (25% v/v in HMDEK buffer). Axonemes were obtained by treating isolated cilia with 1% Igepal CA-630 solution to solubilize 'Membrane + Matrix' fraction.

### Western blotting
Axonemes isolated from *ift46*::NIT IFT46-mNeonGreen, LMJ.RY0402. 233724, and IFT46 mNeonGreen:*VashL* were boiled in 1X Laemmli Buffer and run in equivalents on 4–15% continuous gradient SDS-PAGE gels (Invitrogen). Bands were visualized by Coomassie staining and normalized between samples. To estimate Detyrosinated tubulin,

normalized axoneme equivalents were re-run on 4–15% SDS-PAGE gels, blotted onto PVDF membranes, and probed with anti-Detyrosinated (MAB5566, Merck), Tyrosinated (ABT171, Merck), Polyglycylated ('Glypep-1' from Carsten Janke Lab), Polyglutamylated (GT335, Avantor), Beta Tubulin (SAP) or Acetylated Tubulin antibody (6-11B-1, Santa Cruz), developed using chemiluminescence imager (Azure Biosystems). All antibodies were used at 1:2000 dilution.

To estimate relative enrichment of tyrosinated or detyrosinated tubulin after enzyme treatment, 3 identical treated tubulin batches were run on 4–15% SDS-PAGE gels, blotted onto PVDF membrane. Blots were probed with with either anti-Tyrosinated, Detyrosinated, or Acetylated Tubulin antibodies, developed using chemiluminescence imager (Azure Biosystems).

### Plunge freezing
A Leica Automatic Plunge Freezer EM GP machine was used to perform plunge freezing as it counts with an integrated humidity control chamber. Cu supported 3.5/1 Holey Carbon grids (Quantifoil) were glow discharged on both sides for six seconds each using atmospheric air to render them more hydrophilic. In the meantime, the plunge freezer chamber temperature was set to 32 °C to obtain and maintain high humidity (above 80%). The plunge freezer was set to detect the wetting of the filter paper (Whatman) upon blotting (around 2 seconds blotting time, no automatic plunging), which was mounted on one side only with the purpose of blotting from the grids back side. The sample was loaded on both sides (3.5 µL total) and incubated for 30 seconds before the addition of 1.5 µL of sonicated gold fiducials suspended in PBS, followed by immediate blotting. In many cases automatic blotting was not optimal, requiring swift manual blotting before plunging. Grids were stored in grid-boxes which were themselves stored in cryogenic tanks.

### Cryo-electron tomography
Acquisition of cryo-EM images for wild-type axonemes was done with a 300 kV Thermo Fisher Titan Halo TEM with a field emission gun (FEG) electron source and a Gatan K2 Summit direct electron detector with energy filter using a slit width of 20 eV. Acquisition of cryo-EM images for VashL axonemes was done with a 300 kV Titan Krios and a Falcon4i camera with a Selectris X energy filter with a slit width of 20 eV. To facilitate the location of regions of interest containing well preserved cilia, SerialEM[51] was used to generate a full grid overview by automatically acquiring and stitching low magnification (210x) images. Tilt series were acquired with SerialEM on areas of interest at 30,000x nominal image magnification (33,000x for *VashL* dataset), resulting in a calibrated pixel size of 4.72 Å (counted mode) (3.76 Å for the *VashL* dataset). Tilt series were recorded with 2° increments with a bidirectional tilt scheme from −24° to 42° and from −26° to −42° (3° tilt increment and dose symmetric scheme from 0 to ±60° for the *VashL* dataset. The defocus target varied from −1.5 to −5.5µm and the cumulative dose was 95- 120 e - per Å 2 per tomogram. Images were acquired in the dose fractionation mode with 0.25 s frame time to a total of around 8 frames per tilt image (.eer format for the *VashL* dataset). Drift was kept below 1 nm/s. The frames were aligned using K2Align software (motioncorr2 for the *VashL* dataset), which is based on the MotionCorr algorithm[52]. Tomogram reconstruction was performed using Etomo from IMOD v.4.10.11-α using weighted backprojection[53]. Contrast transfer function curves were estimated with CTFPLOTTER and corrected by phase-flipping with the software CTFPHASEFLIP, both implemented in IMOD[54]. Dose weighted filtering was also performed using the mtffilter command from the IMOD package. In some cases, a nonlinear anisotropic diffusion filter by IMOD[53] was applied on low-contrast tomograms acquired closer to focus. The *VashL* dataset tomograms were filtered with a deconvolution filter.

## Sub-tomogram averaging

IMOD v.4.10.11-α and PEET v1.11.0 were used for sub-tomogram averaging[55]. Using 3dmod, included in the IMOD package, tomograms were inspected looking for signs of adequate structural preservation, namely, roundness and straightness of the axonemes and presence of unaltered axonemal components such as radial spokes and dynein arms. To model the location of each 96-nm repeat within tomograms, the coordinates of points of the central axis of A-tubules at the base of every RS2 were determined manually and saved in.mod format (supported by IMOD software). At least a thousand of these points were determined per experimental condition. Using PEET (v 1.11.0), each sub-tomogram containing a 96-nm repeat was aligned to a common reference so that all microtubule axes remained parallel to each other with the right polarity. The particles from each microtubule doublet were averaged independently to determine the angles of rotation that should be applied to each microtubule doublet to pre-align it with the common reference, providing a good starting point for automated fine alignment. Missing wedge compensation was used, one single particle was commonly used as an initial reference per experiment. Soft masks outlining the 96-nm repeat structure were used for alignment optimization. The maximum search ranges for translations and rotations of each particle during alignment to iteratively refined references were estimated based on the precision achieved during manual modeling and pre-alignment, normally resulting in values of ±12° around the Y axis, ±6° for the remaining axes and 8 pixels for translations in each direction. This strategy, laborious compared to fully automated sub-tomogram averaging routines, provides human- proofread averages without missing-wedge, chirality, or polarity artifacts. Routinely, the final average was generated discarding the worst 10% of particles based on their cross- correlation coefficient score against the final reference (consisting of the average of the 66% best aligned particles). The wild-type dataset consists of 8 flagella (1088 particles) while the *VashL* dataset consists of 6 flagella (840 particles). Visualization of averaged electron density maps was performed in 3dmod from IMOD. 3D rendering of iso-surfaces was performed using UCSF Chimera (v 1.14)[52,56].

## Statistics and reproducibility

All data were plotted and analyzed in Origin Pro2019. The experimental workflows were performed on all conditions end-to-end one after the other on same day, and replicated on separate days in randomized manner. The investigators were not blinded to allocation during experiments and outcome assessments. No data were excluded from analyses. No statistical method was used to predetermine sample size.

## Reporting summary

Further information on research design is available in the Nature Portfolio Reporting Summary linked to this article.

# Data availability

Subtomogram average data for WT or *VashL* 96-nm repeats have been deposited at the Electron Microscopy Databank and are publicly available as of the date of publication. Accession numbers for wild type and *VashL* 96 nm repeats are EMD-50561 and EMD-52022, respectively. Any additional information is available from the lead contacts upon request. Source data are provided with this paper.

# Code availability

A custom Python script was used to compute swimming speeds, beat frequencies and forward and backward displacements per beat cycle, as described previously[57]. The scripts are available from the lead contact upon request.

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

## Acknowledgements

The authors would like to acknowledge Veikko Geyer, Petra Kiesel, Corina Bräuer and the Light Microscopy Facility at MPI-CBG as well as the National Facility for Structural Biology at Human Technopole for technical support. We acknowledge all members of the Pigino and Diez labs for fruitful discussions and Dennis Diener for comments on the manuscript. Tubulin-Tyrosine Ligase was a kind gift from Michel Steinmetz (Paul Scherrer Institut, Switzerland). The project was supported by the Deutsche Forschungsgemeinschaft (DFG, German Research Foundation) under Germany's Excellence Strategy – EXC 2068 – 390729961 – Cluster of Excellence, Physics of Life of TU Dresden (starting grant to A.C), the European Research Council (ERC) under the European Union's Horizon 2020 research and innovation program (grant 819826 to G.P) and the DFG (grant PI1218/3-1 to G.P and SFB 1027 grant to S.D). A.C was further supported by the Dresden International Graduate School for Biomedicine and Bioengineering (DIGS-BB) and A.P.N by an EMBO long–term fellowship (ALTF number 891-2018) as well as by an HFSP cross-disciplinary fellowship (LT000515/2019).

## Author contributions

Project design and supervision: S.D, G.P.; Data acquisition and analysis: A.C., L.S., A.P.N., G.A.V.; Writing-Original Draft: A.C., S.D., G.P.; Writing-Review and Editing: All coauthors.

## Competing interests

The authors declare no competing interests.
