## [Transparent Peer Review file · Nature Communications]

Tubulin tyrosination/detyrosination regulate the affinity and sorting of intraflagellar transport trains on axonemal microtubule doublets

Corresponding Author: Dr Gaia Pigino

Version 0:

Reviewer comments:

Reviewer #1

(Remarks to the Author)

Chhatre et al. reported that lack of detyrosination could lead to the stoppage of IFT trains, higher detachment probability of anterograde trains and lower retrograde trains in VashL *Chlamydomonas*. The authors suggest that the detyrosination of microtubules may lead to bias of tracks for the IFT trains. The stoppage events in the VashL flagella may be caused by the collision of IFT trains.

This study presented innovative ideas including characterization of the detyrosination mutant and the reconstitution assays to analyze the biophysical properties of IFT trains. This manuscript will be of great interests to the general audience in the field of cilia biology, motor proteins and cytoskeleton.

Main points:

The key conclusions would be better supported if statistics of the experiments could be clarified in the main text or figure legends. The reviewer is not sure about how many flagella/axonemes are analyzed for all of the box plots and can't evaluate the quantitative aspects of the conclusion.

One thing to note is that the collision of trains is not directly characterized so it's better to not overstate the results as if the collision happens. It's better to treat it as a hypothesis that is consistent with the data (see specific sentences below).

Line 115:

Fig 1A along shows the stoppages of both anterograde and retrograde trains but does not show collisions between them. An alternative explanation here could be: maybe the trains just experience more stops by themselves without detyrosination.

Line 121:

At this point, the data do not support the collision yet. So this type of statement, plus the following estimation of collision, is better suited for discussion.

Line 135:

It will be great to have some more information about the cryoET imaging:

1. How many tomograms/flagella are imaged/averaged?
2. At what resolution the WT and mutant are indistinguishable?

This information in the main text or legend would make the conclusion in this section more quantitatively supported.

Line 177 and Line 187

The data in this section showed reconstituted trains showed similar biophysical properties to the ones in cells, but no data on the composition of the trains. Saying "All main constitutive components of IFT trains are retained *ex vivo*" and "*ex vivo* reconstitution could recapitulate *in vivo* IFT train composition" are overstatement of the data.

Figure 3D:

What are the total number of events? Please elaborate the different behaviors of the three strains?

Line 237:

“the slowdown of trains observed in VashL cells (Fig. S2B)”-This is the first time this panel is referred to. Calling it a “slowdown” is vague and can be confusing for this section. The speed of trains is actually the same but the average velocity is lower just because of the stoppage events?

Figure S2A:

The CLiP strain was not discussed in the manuscript.

Figure S3:

It is unclear how many flagella were monitored, which is more important to gauge the statistical aspects of the results. Total number of events is different since they could be coming from a limited number of axonemes but for a long time, such results may be caused by axoneme-specific defects. The reviewer has the same question for all other box plots.

Figure 2 is only for the experimental schematics and may be combined with Figure 3?

Line 243:

“In this work we used CRISPR-mediated targeted mutagenesis to show that depletion of detyrosination in cilia impairs IFT motility and logistics, through temporary stalling of IFT trains when two oppositely directed trains cross on the same microtubule.” This is a model consistent with the data. The authors did not “...used CRISPR-mediated targeted mutagenesis to show xxx two oppositely trains cross...”.

Could the authors add some comments about the molecular recognition of the de-tyrosinated microtubules and tyrosinated microtubules with the kinesin and dynein in the discussion?

Could the authors comment about the conservation of the detyrosination enzymes and the relevant mechanism across species? Or even within the same species, would the same mechanisms be used for trachea cilia and sperm flagella?

Reviewer #2

(Remarks to the Author)

This manuscript entitled “Tubulin tyrosination/detyrosination regulates the sorting of intraflagellar transport trains on axonemal microtubule doublets” by Chhatre et al describes the effects of tyrosination/detyrosination cycle in intraflagellar transport. The study uses CRISPR to K/O detyrosinase enzyme and combines live imaging and in vitro reconstitution to dissect the effects on anterograde and retrograde transport. Overall, the manuscript data is of high quality and well written, which will be of interest to the microtubule cytoskeleton and molecular motor community. I have few concerns and comments regarding the data interpretation and the model presented here. In light of this, I do not recommend publication in its current form and a significant revision is necessary.

Some specific queries are outlined below;

1. The CRISPR KO of VASH leads the decreased detyrosinated levels, which is clear from the blots presented in Figure S2. However, it is well-known in the field that Tyr/DeTyr levels affect other PTMs including glutamylation and glycylation. Therefore, it is imperative to check the glutamylation and glycylation levels before concluding that the phenotypic effects seen or not due to pleiotropic effects of other PTMs.
2. The IFT train stoppages seen are mostly at the mid-way of the flagella, in the model presented in the Figure 6 it is proposed that the B-tubule is predominantly detyrosinated and in the VASH K/O both the tubules are tyrosinated, which affects the sorting of IFT trains. It is not clear how the authors derived the differential Tyr/DeTyr modification between A- and B-tubules.
3. From the IFT motility data the authors claim that the stoppage is due to the collision events between missorted anterograde IFT trains. However, there are many IFT trains that are also stalled without any crossover trains, it will be ideal to segregate these crossover vs non-crossover events and quantify the stalling IFT trains.
4. On similar lines, how did the authors rule out whether the stalling of anterograde trains is due to switch between heterotrimeric and homodimeric kinesin-2 motors. It is possible that the PTM states affect the switching between two types of anterograde kinesins.
5. In the ex-vivo experiments the authors have used TTL or CPA treated porcine brain microtubules to generate tyrosinated or detyrosinated microtubules respectively. While it is generally accepted to use CPA treated brain microtubules for homogenous detyrosinated microtubules. However, it is not possible to generate tyrosinated microtubules from brain microtubules by only TTL treatment and it well-documented that tyrosinated tubulin/microtubules can be prepared from HEK or HeLa cells. Therefore, the data presented in Figure S9 does not truly reflect Tyr/DeTyr microtubules used for ex-vivo experiments.
6. In general, the manuscript will benefit by describing the VASH K/O *Chlamydomonas* motility flagella beating, length and IFT phenotype, in line with data presented in Figure S2 and point no. 1 in this reviewer comment.

Version 1:

Reviewer comments:

Reviewer #1

(Remarks to the Author)

The authors did an excellent job addressing the comments and strengthening the quantitative analyses of their experiments.

A minor suggestion:

“These stoppage events predominantly correlated with the crossings of opposite-polarity trains (Fig. 1A, white 121 arrowheads) and virtually no non-crossover related stoppages were observed during the movie recordings, suggesting that in VashL cells anterograde and retrograde trains could experience collisions (see Discussion).”

This is such an important experiment/conclusion and it would be great to clarify the indications. “predominantly” and “virtually” are a bit vague. Could the authors developed some more quantitative assessment? Like having the opposite train within certain distance and time range, how many events fall into those ranges?

Reviewer #2

(Remarks to the Author)

The authors have sufficiently addressed previous queries, I have no further concerns except for few minor points in the discussion.

1. Page 10, lines 293-297; IFT models have been described in other systems not restricted to *C. elegans*, authors should generalize this statement.
2. Page 11, line 319; authors should clearly state about detyrosination/tyrosination modification/
3. Page 11, line 322; There was no yeast kinesin-2 used in Sirajuddin et al 2014 study.

We thank all reviewers for their careful and constructive review of our manuscript. Please find below our detailed point-to-point responses to the reviewers' comments and our actions taken. In addition, the key changes are also indicated by blue color in a submitted version of the manuscript for review only.

We (i) performed additional experiments, (ii) supplied clarifications and (iii) extended the interpretation and discussion of our results (at various places in the manuscript).

Reviewer #1 (Remarks to the Author):

Chhatre et al. reported that lack of deetyrosination could lead to the stoppage of IFT trains, higher detachment probability of anterograde trains and lower retrograde trains in VashL *Chlamydomonas*. The authors suggest that the deetyrosination of microtubules may lead to bias of tracks for the IFT trains. The stoppage events in the VashL flagella may be caused by the collision of IFT trains.

This study presented innovative ideas including characterization of the deetyrosination mutant and the reconstitution assays to analyze the biophysical properties of IFT trains. This manuscript will be of great interests to the general audience in the field of cilia biology, motor proteins and cytoskeleton.

Main points:

The key conclusions would be better supported if statistics of the experiments could be clarified in the main text or figure legends. The reviewer is not sure about how many flagella/axonemes are analyzed for all of the box plots and can't evaluate the quantitative aspects of the conclusion.

Response: We apologize for this oversight and thank the reviewer for pointing it out. **We have now clearly mentioned total number of trains, number of cells/flagella, statistical testing and experimental replicates in figure captions wherever applicable.**

One thing to note is that the collision of trains is not directly characterized so it's better to not overstate the results as if the collision happens. It's better to treat it as a hypothesis that is consistent with the data (see specific sentences below).

Line 115: Fig 1A along shows the stoppages of both anterograde and retrograde trains but does not show collisions between them. An alternative explanation here could be: maybe the trains just experience more stops by themselves without deetyrosination.

Response: Virtually no non-crossover related stoppages were seen for the duration of recording the movies. **We now clarify this in the text.**

Line 121: At this point, the data do not support the collision yet. So this type of statement, plus the following estimation of collision, is better suited for discussion.

Response: **We have reworded the results section and also the end of Introduction referring 'crossings' and 'stoppages', whereas 'collisions' are only speculated. The estimation and interpretation of collisions is shifted to Discussion upon reviewer's suggestion.**

Line 135: It will be great to have some more information about the cryoET imaging: How many tomograms/flagella are imaged/averaged?

Response: The wild-type dataset consists of 8 flagella (1088 particles) while the VashL KO dataset consists of 6 flagella (840 particles). **We now mention this in Methods section.**

At what resolution the WT and mutant are indistinguishable? This information in the main text or legend would make the conclusion in this section more quantitatively supported.

Response: Based on the reported FSC curves, the models are indistinguishable up to 30Å resolution. **We mention this information in the section of the text related to the cryo-EM results.**

Line 177 and Line 187: The data in this section showed reconstituted trains showed similar biophysical properties to the ones in cells, but no data on the composition of the trains. Saying “All main constitutive components of IFT trains are retained *ex vivo*” and “*ex vivo* reconstitution could recapitulate *in vivo* IFT train composition” are overstatement of the data.

Response: We agree with the reviewer that our direct data on the composition of trains *ex vivo* is limited to only a three IFT proteins. However, among all IFT proteins we specifically labeled IFT46, IFT140, and Light Chain of Dynein 1b that are known to be stably associated with their respective sub-complexes (IFT-A, IFT-B, IFT Dynein, respectively) even after extraction from cells. We believe that consistent observation of fast processive runs for all these IFT components are indicative of conservation of IFT train compositional and functional properties *ex vivo*. **Agreeing with the reviewer about the overstatement, we have reworded this section and cited additional publications (from Esben Lorentzen, Doug Cole) which talk about aforementioned association of IFT sub complexes.**

Figure 3D: What are the total number of events? Please elaborate the different behaviors of the three strains?

Response: **We have edited in the total events counted for each case for Fig 3D.**

Line 237: “the slowdown of trains observed in *VashL* cells (Fig. S2B)”-This is the first time this panel is referred to. Calling it a “slowdown” is vague and can be confusing for this section. The speed of trains is actually the same but the average velocity is lower just because of the stoppage events?

Response: We thank the reviewer for pointing out the confusing wording. We have recalculated the *in vivo* WT and *VashL* train velocities excluding the stoppages. Consistent with our previous claim (and as the reviewer correctly predicts) we observe that the speeds are unaffected. Additionally *in vivo* speeds match *ex vivo* speeds, consistent with our claim that Tubulin Tyrosination/Detyrosination does not affect train velocities. **We have essentially combined *in vivo* and *ex vivo* velocity plots into the new Figure S9C, and removed previous Figure S2b for clarity.**

Figure S2A: The CLiP strain was not discussed in the manuscript.

Response: **We apologize for missing out on mentioning the CLiP mutant and talk about it in results.**

Figure S3: It is unclear how many flagella were monitored, which is more important to gauge the statistical aspects of the results. Total number of events is different since they could be coming from a limited number of axonemes but for a long time, such results may be caused by axoneme-specific defects. The reviewer has the same question for all other box plots.

Response: **We have now clearly mentioned total number of trains, number of cells/flagella, statistical testing and experimental replicates in figure captions wherever applicable.**

Figure 2 is only for the experimental schematics and may be combined with Figure 3?

Response: That is a reasonable suggestion. Although, as the method to reconstitute native IFT motility contributes to the novelty of this work, we would prefer keeping the experimental schematic (Figure 2) and relevant Supplementals (S5, S6 etc.) separate from actual *ex vivo* data (Figure 3) as modular reference to

readers. Combining the two will make the new Figure a bit cluttered and heavy. **We have edited the references to the figure in the main text.**

Line 243: “In this work we used CRISPR-mediated targeted mutagenesis to show that depletion of detyrosination in cilia impairs IFT motility and logistics, through temporary stalling of IFT trains when two oppositely directed trains cross on the same microtubule.” This is a model consistent with the data. The authors did not “...used CRISPR-mediated targeted mutagenesis to show xxx two oppositely trains cross...”.

Response: We have reworded these sentences to say that “In this work we used CRISPR-mediated targeted mutagenesis to investigate the effect of depletion of tubulin detyrosination on IFT motility and logistics...”, further providing interpretations on collisions later in the Discussion.

Could the authors add some comments about the molecular recognition of the detyrosinated microtubules and tyrosinated microtubules with the kinesin and dynein in the discussion?

Response: We now refer to single-molecule studies of the effect of Tyr/Detyr PTMs on Kinesin and Dynein in the Discussion, and also some known candidates with known tubulin sensing moieties.

Could the authors comment about the conservation of the detyrosination enzymes and the relevant mechanism across species? Or even within the same species, would the same mechanisms be used for trachea cilia and sperm flagella?

Response: Nice suggestion. Detailed literature on mechanistic insight and conservation of detyrosination enzymes is sparse, owing to their relatively recent discovery, although few recent studies indicate its role in different models. Same for organ/organelle specific regulation. **We have now added a description on this topic in the Discussion citing relevant literature and our speculations.**

Reviewer #2 (Remarks to the Author):

This manuscript entitled “Tubulin tyrosination/detyrosination regulates the sorting of intraflagellar transport trains on axonemal microtubule doublets” by Chhatre et al describes the effects of tyrosination/detyrosination cycle in intraflagellar transport. The study uses CRISPR to K/O detyrosinase enzyme and combines live imaging and in vitro reconstitution to dissect the effects on anterograde and retrograde transport. Overall, the manuscript data is of high quality and well written, which will be of interest to the microtubule cytoskeleton and molecular motor community. I have few concerns and comments regarding the data interpretation and the model presented here. In light of this, I do not recommend publication in its current form and a significant revision is necessary.

Some specific queries are outlined below;

The CRISPR KO of VASH leads to the decreased detyrosinated levels, which is clear from the blots presented in Figure S2. However, it is well known in the field that Tyr/DeTyr levels affect other PTMs including glutamylation and glycylation. Therefore, it is imperative to check the glutamylation and glycylation levels before concluding that the phenotypic effects are seen or not due to pleiotropic effects of other PTMs.

Response: We have now performed Western Blots against other PTMs (acetylation, polyglycylation and polyglutamylation). We found no discernible differences between WT or VashL axonemes. **We add these Results in Figure S2 and the corresponding antibodies used in Materials.**

The IFT train stoppages seen are mostly at the mid-way of the flagella, in the model presented in the Figure 6 it is proposed that the B-tubule is predominantly detyrosinated and in the VASH K/O both the tubules are tyrosinated, which affects the sorting of IFT trains. It is not clear how the authors derived the differential Tyr/DeTyr modification between A- and Btubules.

Response: Strong biochemical and immunoEM evidence from Joe Howard, Karl Lehtreck, Stefan Geimer and Karl Johnson shows that B-tubules are detyrosinated, and A-tubules are tyrosinated. **We now highlight this fact in the Discussion on the model of train sorting and cite the literature from these labs. We now also cite Lehtreck and Geimer at other instances where Tyr/DeTyr asymmetry is mentioned.**

From the IFT motility data the authors claim that the stoppage is due to the collision events between missorted anterograde IFT trains. However, there are many IFT trains that are also stalled without any crossover trains, it will be ideal to segregate these crossover vs non-crossover events and quantify the stalling IFT trains.

Response: Virtually no non-crossover related stoppages were seen for the duration of recording the movies. **We now mention this in the text.**

On similar lines, how did the authors rule out whether the stalling of anterograde trains is due to switch between heterotrimeric and homodimeric kinesin-2 motors. It is possible that the PTM states affect the switching between two types of anterograde kinesins.

Response: Chlamydomonas does not have homodimeric IFT kinesins. Discreet evidence for sorting by detyrosination in two-kinesin IFT system is lacking. Minhajuddin *et. Al.* 2014 show increased processivity of Kinesin-2 on detyrosinated microtubules, **which we refer in the text.** Since homodimeric kinesins belong to the same family as heterotrimeric kinesins, they could use similar sorting mechanism in relevant models such as *C.elegans*. We agree with the reviewer that it would be interesting to study if PTM states affects cargo hand-off between two types of anterograde kinesins. **We believe this is beyond the scope to of current study but recognize its relevance, and elaborate the point in the Discussion.**

1 In the ex-vivo experiments the authors have used TTL or CPA treated porcine brain microtubules to generate tyrosinated or detyrosinated microtubules respectively. While it is generally accepted to use CPA treated brain microtubules for homogenous detyrosinated microtubules. However, it is not possible to generate tyrosinated microtubules from brain microtubules by only TTL treatment and it is well-documented that tyrosinated tubulin/microtubules can be prepared from HEK or HeLa cells. Therefore, the data presented in Figure S9 does not truly reflect Tyr/DeTyr microtubules used for ex-vivo experiments.

Response: We acknowledge the reviewer's comment essentially regarding the complexity of tubulin variants present in brain tubulin, such as $\Delta 2$ and $\Delta 3$. While TTL treatment does not produce 100% tyrosinated tubulin, our data consistently show that the relativistic depletion or enrichment of tyrosinated tubulin affects train landing dynamics. This supports our interpretation that anterograde trains preferentially interact with detyrosinated tubulin, whereas retrograde trains favor tyrosinated tubulin. This finding aligns with our overall model that suggests train affinity influences *in vivo* stoppage-free runs. The role of aforementioned tubulin variants and the enzymes in regulating IFT and cilia function is interesting question to address for future.

Regarding Figure S9, the tubulin used was sourced from the same purification batch, treated with TTL or CPA, and then equally split for Western blot normalization and ex vivo motility assays. We have clarified this procedure in the Methods section under "Enzyme treatment of microtubules."

In general, the manuscript will benefit by describing the VASH K/O *Chlamydomonas* motility, flagella beating, length and IFT phenotype, in line with data presented in Figure S2 and point no.1 in this reviewer comment.

Response: **We added the steady state flagella length. Beat frequency, Cell displacement per beat and swimming speed in Supplemental Figure S2 and now mention the result in the relevant section.** Regarding the IFT phenotype: There is no evidence to indicate that Vash-SVBP detyrosinates any IFT complex proteins directly. Further, previous work from Steinmetz Lab (DOI: 10.1083/jcb.202205096) shows that the complex preferentially binds to microtubule lattice to catalyze cleaving of the Tyrosine. **We discuss this clearly in the Discussion section.**

We thank both reviewers for their encouraging responses to our revised manuscript.

Reviewer #1 (Remarks to the Author):

The authors did an excellent job addressing the comments and strengthening the quantitative analyses of their experiments.

A minor suggestion:

“These stoppage events predominantly correlated with the crossings of opposite-polarity trains (Fig. 1A, white 121 arrowheads) and virtually no non-crossover related stoppages were observed during the movie recordings, suggesting that in VashL cells anterograde and retrograde trains could experience collisions (see Discussion).”

This is such an important experiment/conclusion and it would be great to clarify the indications. “predominantly” and “virtually” are a bit vague. Could the authors developed some more quantitative assessment? Like having the opposite train within certain distance and time range, how many events fall into those ranges?

We carefully checked our data again. In consequence, we were able to change the wording to "All observed stoppage events coincided with crossing events of opposite-polarity trains (white arrowheads) and no stoppage events were detected at times or locations where crossing events were absent. Given that observed crossing events indicate that the respective trains rarely travelled on the same microtubule doublet, this suggests that in the VashL cells, anterograde and retrograde trains frequently collided when sharing the same doublet, leading to temporary train stoppages (see also Discussion). "

Reviewer #2 (Remarks to the Author):

The authors have sufficiently addressed previous queries, I have no further concerns except for few minor points in the discussion.

1. Page 10, lines 293-297; IFT models have been described in other systems not restricted to *C. elegans*, authors should generalize this statement.

Fixed.

2. Page 11, line 319; authors should clearly state about deetyrosination/tyrosination modification/

Fixed.

3. Page 11, line 322; There was no yeast kinesin-2 used in Sirajuddin et al 2014 study.

Apologies for the oversight. We edited the statements accordingly.